# Diagnostic accuracy of computed tomography in adults with suspected acute appendicitis at the emergency department in a private tertiary hospital in Tanzania

**Masawa K. Nyamuryekung'e**[ID]*, **Miten R. Patel**[○], **Ahmed Jusabani**[○], **Ali A. Zehri**[○], **Athar Ali**[○]

Department of Surgery, The Aga Khan Hospital, Dar es salaam, Tanzania

○ These authors contributed equally to this work.
* masawa4@hotmail.com

**Data Availability Statement:** The data in the study is the property of AKU, Dar Campus (third party); The data points can only be available on request to

## Abstract

### Introduction

The increasing incidence of acute appendicitis in sub-Saharan Africa emphasizes the need for accurate and reliable diagnostic tools. However, the variability in the diagnostic performance of computed tomography for suspected acute appendicitis coupled with comparatively higher negative appendectomy rates in this setting highlight a possible concern regarding the diagnostic accuracy.

This study evaluated the diagnostic accuracy of a computed tomography scan for suspected acute appendicitis at the emergency department in Tanzania.

### Methods

A retrospective diagnostic accuracy study was conducted from July to October 2020. All patients above 14 years of age who presented at the emergency department with right iliac fossa abdominal pain of fewer than ten days and underwent computed tomography for suspected acute appendicitis were evaluated, and the Alvarado score was computed. Histological diagnosis and clinical follow-up of 14 days were considered the reference standard. Ethical clearance was sought from the Aga Khan University Ethical review committee.

### Results

176 patients were included in this study. The sensitivity, specificity, and diagnostic accuracy were 100% (95% CI 91.8–100), 96.9% (95% CI 92.2–99.1), and 96.9% (95% CI 93.1–98.3), respectively. The mean Alvarado score in those without acute appendicitis was 4 (95% CI 3.7–4.3) compared to a mean score of 6.6 (95% CI 6.0–7.2) amongst those with acute appendicitis. The area under the receiver operator characteristics curve of computed tomography was 98.4%, and that of the Alvarado score was 84.1%.

Agha Khan University, The Research Administrative office, Ms. Mwanaarab R. Sibuma, The Aga Khan University Medical College, P.O. Box 38129, Dar es salaam Tanzania, Cell: +255 682 00097, e-mail: mwanaarab.sibuma@aku.edu.

**Funding:** The authors received no funding for this work.

**Competing interests:** The authors have declared that no competing interests exist.

**Abbreviations:** AA, Acute Appendicitis; AS, Alvarado Score; AUROC, Area Under the Receiver Operator Characteristics curve; CT, Computed Tomography; CI, Confidence interval; EMD, Emergency Medicine Department; LR+, Positive Likelihood ratio; LR, Negative Likelihood ratio; ROC, Receiver operator characteristics; SSA, Sub Saharan Africa; WSES, World society of emergency surgery.

## Conclusions

The diagnostic performance of computed tomography in this study is similar to that established elsewhere. However, the Alvarado score is not routinely used for the initial screening of suspected acute appendicitis patients. A threshold of Alvarado score of 4 as a guide to conduct computed tomography for suspected acute appendicitis would have decreased computed tomography use by 50%, and missed 4 cases. Implementation studies that address Alvarado score use should be conducted.

## Introduction

Acute appendicitis (AA) is the commonest cause of acute abdomen in adults that frequently requires emergency surgical intervention [1]. AA has been viewed primarily as a western disease; however, current disease trends have suggested increasing incidence in sub-Saharan Africa (SSA), underscoring its growing significance [2].

An accurate diagnosis of AA can be made clinically about 50% of the time. In the remaining instances, diagnosis solely through clinical findings may be challenging [3]. There is an increased likelihood of diagnosis delays or misdiagnosis in this scenario. This may result in higher appendicular perforation and negative appendectomy rates [4,5]. To improve the diagnostic accuracy of AA, medical imaging tools are frequently utilized in addition to clinical findings. The abdominal computed tomography (CT) is the commonest preferred medical imaging used for this purpose in nonpregnant adults [6]. It allows for visualization of the appendix, evaluation of features known to predict AA and determining alternative pathologies [7,8].

Multiple systematic reviews and meta-analyses have established that CT's diagnostic accuracy for suspected AA in nonpregnant adult patients is over 92% [7–12]. However, in the pooled meta-analysis studies, the range of sensitivity and specificity of CT's diagnosis for suspected AA was 72 to 100% and 50 to 100%, respectively [12]. Variability of diagnostic accuracy may be due to radiologists' experience, contrast use, differing CT generations, imaging protocols, and acute appendicitis prevalence in the study settings [7,9,11,12].

There have been over ten meta-analyses and systematic reviews conducted to date assessing the diagnostic performance of CT scans for suspected AA in adults, and none include findings from SSA, possibly due to the lack of eligible studies from this area [7–15]. This partly reflects the limited resources of health facilities in terms of equipment available, expertise, experience, and possibly the preference of the imaging modality for suspected AA [16–19]. The outcomes of performed appendectomies in SSA reveal a negative appendectomy rate of about 17%, compared to 1–5% in other areas, raising concern for diagnostic accuracy of suspected AA in this region [20,21].

The current study's main objectives were to determine the sensitivity, specificity, diagnostic accuracy, and predictive values of CT for patients with suspected AA presenting to the emergency medicine department (EMD) of Aga Khan Hospital, Tanzania.

## Methods

Ethical clearance for the study protocol was sought from the Aga Khan University Ethical review committee, AKU-ERC, EA, with a reference number of AKU/2020/0170/fb. Consent was waived as this was a retrospective chart review and the data collected was de identified.

The study was conducted from July to October 2020 at a private tertiary teaching hospital in Dar es Salaam, Tanzania. Records of patients aged 14 years and above who presented at the EMD with right iliac fossa colicky abdominal pain for less than 10 days and underwent CT abdomen for suspected AA were included in this study. Pregnant women and those who underwent abdominal-pelvic ultrasonography prior to the abdominal CT scan were excluded. Sample size estimation based on the Buderer formula of diagnostic accuracy for unknown disease prevalence was employed [22]. The assumptions applied were an acceptable marginal error of 0.05, hypothesized sensitivity of 0.95, specificity of 0.94 and prevalence of 0.43, based on Bo Rud, 2019 et al. [12]. The required sample size for sensitivity and specificity were 170 and 153 participants, respectively. Consecutive sampling was utilized to include patient records into the study until the sample size was reached.

The index test for suspected AA was a helical CT scan manufactured by Philips, Ingenuity model, 128 slices with 64 detector rows. One of four consultant medical radiologists interpreted the images individually during their on-call schedule. All radiologists had at least trained as Master of Medicine in medical radiology. The target condition was AA, encompassing both simple and complicated presentations. All cases diagnosed by CT to have simple or complicated AA were considered test positive. Alternately, all cases with a normal appendix on CT evaluation or alternative diagnoses were considered test negative. Equivocal cases on CT were also captured. Results of other additional tests performed, such as complete blood count (CBC) or urinalysis that were performed were captured, along with clinical findings that were used to calculate individual patient's Alvarado scores.

Histological diagnoses for those who underwent appendectomy and a follow up of two weeks from the initial evaluation in outpatient clinic were considered the reference standard. Reference standard positive was defined histologically as the appendix with transmural presence of acute inflammatory cells. The reference standard negative was defined as specimens not fitting the histological definition for AA in those who underwent appendectomy or did not require an appendectomy after two weeks follow up.

Those with equivocal findings on the index test or those that did not undergo the reference standard were excluded in diagnostic performance analysis. Sensitivity, specificity, diagnostic accuracy, predictive values, and area under the receiver operator characteristics curve (AUROC) were determined for the index test of suspected AA. Analyses were conducted using the IBM SPSS version 25 Statistical package.

## Results

A total of 176 patient records were included in the study, of which 29.5% (52/176) were females, and the mean age was 35 years (95% CI of 34–37). The vast majority (91.5%) had no pre-existing comorbidities. The mean duration of illness, defined as number of days from onset of abdominal pain to emergency medicine department (EMD) presentation was 2 days (95% CI 1.8–2.3). At the EMD, the first assessor was a medical officer, defined as registered medical doctor with Doctor of Medicine degree qualification only, in 40.3%, surgical residents 36.4%, and a general surgery specialist in 23.3% of the cases.

All patients were assessed clinically and underwent a CBC. The mean computed AS was 4.64 (95% CI 4.32–4.97). Those with an AS of 1 to 4 were 50.6%, 5 to 8 were 43.2%, and greater than 8 were 6.3%. Urinalysis was done in 134 patients and was normal in 79.1% of these, 19.4% showing microscopic hematuria, and pyuria with leucocytes in 1.5%. The distribution of the demographic features is summarized in Table 1.

All participants underwent CT investigation for suspected AA. Plain CTs were done in 87.5% of the patients, 10.2%, 1.7%, and 0.6% used intravenous, oral and rectal contrast,

**Table 1. Distribution of participants demographic features (N = 176).**

|  | Frequency | Percentage (%) |
|---|---|---|
| **Sex of Participants** | | |
| **Males** | 124 | 70.5 |
| **Female** | 52 | 29.5 |
| **Age Group in Years** | | |
| **<25** | 31 | 17.6 |
| **26–35** | 59 | 33.5 |
| **36–45** | 56 | 31.8 |
| **>46** | 30 | 17.0 |
| **Duration of Illness in Days** | | |
| **1** | 105 | 59.7 |
| **2** | 24 | 13.6 |
| **3** | 17 | 9.7 |
| **4** | 11 | 6.3 |
| **>5** | 19 | 10.8 |
| **Alvarado Score groups** | | |
| **1 to 4** | 89 | 50.6 |
| **5 to 8** | 76 | 43.2 |
| **>8** | 11 | 6.3 |
| **Comorbid** | | |
| **Present** | 15 | 8.5 |
| **Absent** | 161 | 91.5 |
| **Urinalysis** | | |
| **Abnormal** | 28 | 15.9 |
| **Normal** | 106 | 60.2 |
| **Not done** | 42 | 23.9 |
| **Designation of Assessor** | | |
| **Registrar** | 71 | 40.3 |
| **Resident** | 64 | 36.4 |
| **Consultant** | 41 | 23.3 |

respectively. On CT scan AA was confirmed in 26.7% (47/176) of the cases, of which 17.0% (8/47) were complicated. Of those determined to have AA on CT and underwent appendectomy, 43 were confirmed histologically to have AA, while 4 were not.

The CT scan results were negative for AA in 70.4% (124/176). Of these, 77.4% (96/124) had alternative diagnoses. The commonest alternative diagnosis was urolithiasis in 70.8% (68/96). Of the remaining CT scan negative cases for AA without alternative diagnoses, 3 of 28 participants underwent appendectomy following the observation period and were histologically negative. The other 25 participants completed the 14-day outpatient follow-up and were declared not to have AA. The flow of participants is summarized in Fig 1 below.

To assess which features were associated with CT diagnosis of AA, binomial logistic regression analysis was done whereby calculated AS had a statistically significant association with CT diagnosis of AA. The mean AS in those without CT confirmed AA was 4 (95% CI 3.7–4.3) compared to mean AS of 6.6 (95% CI 6.0–7.2) among those with CT confirmed AA. In contrast, sex, age, urinalysis results or designation of the assessor did not have a statistically significant association.

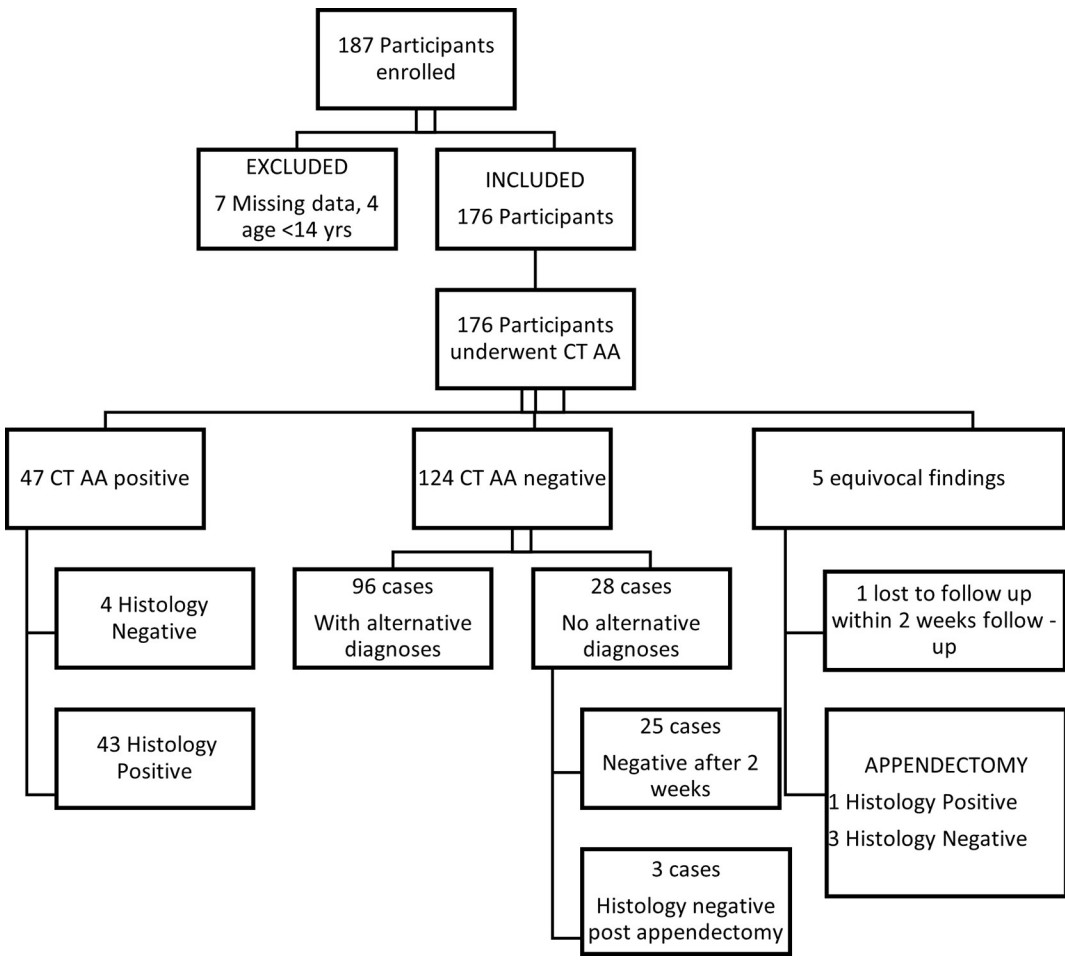

**Fig 1. Flow of participants.**

The prevalence of AA in this study was 25.1% (44/175). For diagnostic performance analysis, 171 participants were included, as 5 had equivocal findings. Cross-tabulation between CT and the reference standards is summarized in Table 2. The sensitivity, specificity, and diagnostic accuracy of CT for adults with suspected AA were 100% (95% CI 91.8–100), 96.9% (95% CI 92.2–99.1), and 96.9% (95% CI 93.1–98.3), respectively. Thus, at a prevalence of 25% in our study, the CT's positive and negative predictive values were 91.5% (95% CI 80.1–96.6) and 100% (95% CI 97.0–100).

ROC curve for the CT revealed an AUROC of 98.4%, and that of AS had an AUROC of 84.1%. A cut-off AS value of 4 had a sensitivity of 90.7% (95% CI 77–97), specificity of 65.6% (95% CI 56.7.1–73.7) and an NPV of 95.4% (95% CI 89–98), and was selected as a cut off to

**Table 2. Cross-tabulation between CT results and reference standard.**

|  | REFERENCE STANDARD | | |
|---|---|---|---|
|  | **Negative** | **Positive** |  |
| **Index Test Negative** | 124 | 0 | 124 |
| **Index Test Positive** | 4 | 43 | 47 |
| **Total** | 128 | 43 | 171 |

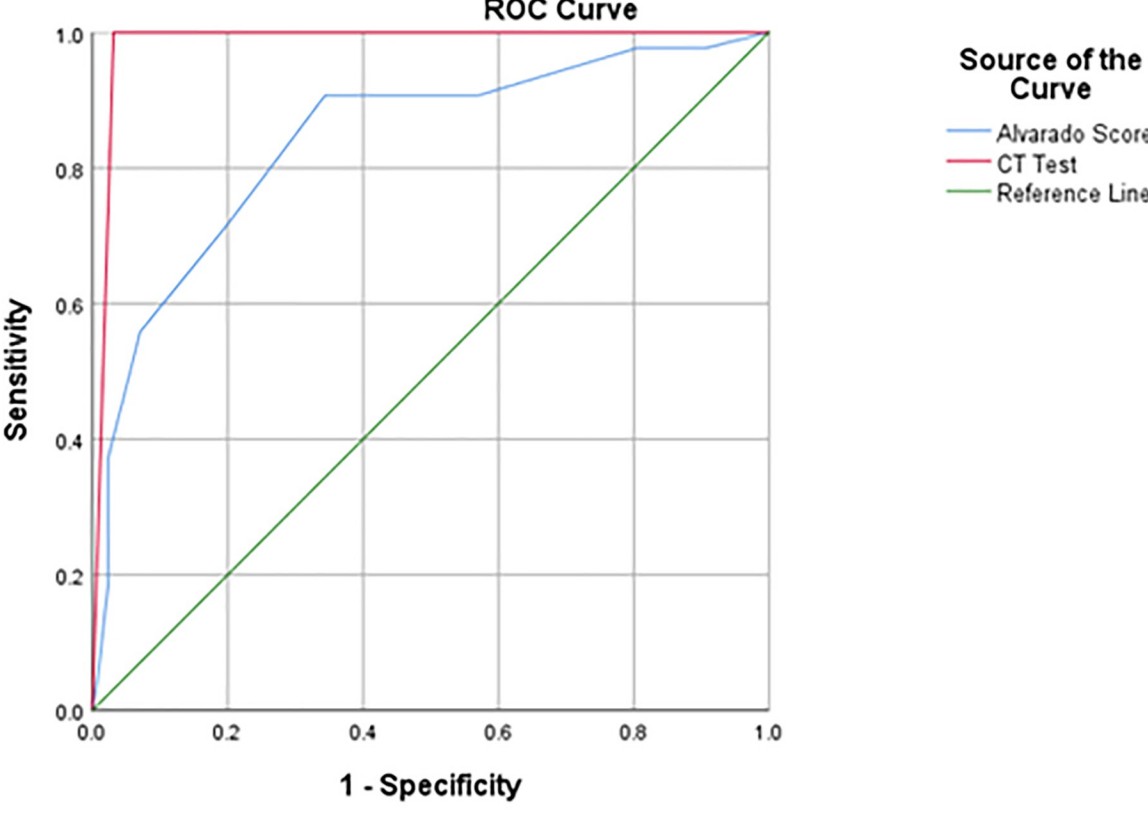

**Fig 2. Receiver operator characteristics curve of CT and Alvarado score.**

rule out AA. An AS of 8 had a specificity of 97% (95% CI 93–99) and a PPV of 84% (95% CI 61–94).

The ROC curves are shown in Fig 2.

## Discussion

The contextual differences in SSA had led to the hypothesis that the diagnostic accuracy of CT for AA might be different. However, in this study, the diagnostic accuracy found is comparable to that demonstrated in metanalyses elsewhere [7–12]. The excellent diagnostic performance of CT for suspected AA needs to be considered in the context of healthcare costs, length of stay at the EMD and adverse risk of radiation exposure [23,24].

A large retrospective study evaluating the diagnostic accuracy of CT for suspected AA identified 106 patients who could have avoided CT by using AS for screening. The collective duration spent at the EMD from ordering to receiving the CT report for these patients was 10 239 minutes [23]. In another study over 5 years, the cumulative cost was 173,998 USDs for CT investigation that could have been avoided in males younger than 45 years presenting with classical features of AA [24]. In the current study, 70% (124/176) of the CT investigations for suspected AA were negative, raising the concern of whether this rate could have been lowered. An AS of 4 or less has sufficient sensitivity to rule out AA as concluded by international guidelines [25,26]. Using this AS cut-off as a threshold to conduct CT, 50% of patients could have avoided CT investigations and missed 4 cases of AA. There are no internationally agreed AS

cut off to proceed to appendectomy, in this study using solely a cut off AS of 8 to conduct appendectomy without CT evaluation would have resulted in a negative appendectomy rate of 15% (3/19) that is considered unacceptable in the CT use era [21].

The AS was not documented in patients' charts in this study and was computed based on the collected data. Whether this resulted from attitudes and practices towards AS use for suspected AA could not be determined. However, it has been previously shown that AS is used in only 6% of suspected AA in this setting [27]. Reasons for its uncommon use need to be determined and addressed to decrease CT overuse and negative appendectomy rates. To improve the cost-effective use of CT, we further emphasize the use of AS for screening all patients with suspected AA.

This study has some limitations; being a retrospective study, it relied on accurate documentation by the attending physicians and radiologists. Four independent radiologists interpreted the CT scans raising concern for inter-observer variability and there is a risk of inclusion bias. The prevalence of AA in our study was 25%, instead of our sample size assumption of 43%. This lowered the precision of our results to 0.067, as opposed to our initial precision estimate of 0.05. At this precision of 0.067, the minimum sample size required is 163 participants.

## Conclusion and recommendations

Overall, the diagnostic performance of CT demonstrated in this study is like that established outside of SSA, and the prevailing issue should be its selective use without adversely affecting AA outcomes. The use of AS as a screening tool would increase the cost-effective use of CT. An AS of 4 should be used as a threshold to conduct CT. Implementation studies that would address the low use of AS should be conducted.

## Supporting information

**S1 Checklist.**
(DOCX)

## Acknowledgments

We thank Kasusu Klint Nyamuryekung'e (DDS, MPhil Int. Health, PhD) for his assistance and inputs in the statistical analysis and preparation of the manuscript.

## Author Contributions

**Conceptualization:** Masawa K. Nyamuryekung'e, Ali A. Zehri, Athar Ali.

**Data curation:** Masawa K. Nyamuryekung'e, Miten R. Patel.

**Formal analysis:** Masawa K. Nyamuryekung'e, Ahmed Jusabani.

**Methodology:** Masawa K. Nyamuryekung'e.

**Project administration:** Masawa K. Nyamuryekung'e.

**Supervision:** Miten R. Patel, Ali A. Zehri, Athar Ali.

**Validation:** Ahmed Jusabani, Ali A. Zehri, Athar Ali.

**Writing – original draft:** Masawa K. Nyamuryekung'e.

**Writing – review & editing:** Miten R. Patel, Ahmed Jusabani, Ali A. Zehri, Athar Ali.

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
