## [Decision Letter · Decision Letter 0]

18 May 2021

PONE-D-21-13993

DIAGNOSTIC ACCURACY OF COMPUTED TOMOGRAPHY IN ADULTS WITH SUSPECTED ACUTE APPENDICITIS AT THE EMERGENCY DEPARTMENT IN PRIVATE TERTIARY HOSPITAL IN TANZANIA

PLOS ONE

Dear Dr. Nyamuryekung'e,

Thank you for submitting your manuscript to PLOS ONE. After careful consideration, we feel that it has merit but does not fully meet PLOS ONE’s publication criteria as it currently stands. Therefore, we invite you to submit a revised version of the manuscript that addresses the points raised during the review process.

Please address the issues and revise accordingly.

We look forward to receiving your revised manuscript.

Kind regards,

Academic Editor

PLOS ONE

Journal Requirements:

Reviewers' comments:

Reviewer's Responses to Questions

**Comments to the Author**

1. Is the manuscript technically sound, and do the data support the conclusions?

Reviewer #1: Partly

Reviewer #2: Yes

2. Has the statistical analysis been performed appropriately and rigorously? 

Reviewer #1: No

Reviewer #2: No

3. Have the authors made all data underlying the findings in their manuscript fully available?

Reviewer #1: Yes

Reviewer #2: No

4. Is the manuscript presented in an intelligible fashion and written in standard English?

Reviewer #1: Yes

Reviewer #2: No

5. Review Comments to the Author

Reviewer #1: The authors aimed to study the diagnostic accuracy of acute appendicitis (AA) by computed tomography (CT) in Tanzania. They recruited 180 patients suspected AA with CT retrospectively in a single hospital. The reference standards were histological examination of the removed appendix as well as clinical follow‐up for 14 days of participants who did not have surgery. After excluded 7 patients with negative CT and loss of follow-up, 173 participants were included in analysis of which 45 had and 128 did not have AA. The CT's diagnostic accuracy for AA was 97.5%, sensitivity and specificity were 91.6% and 100% respectively. The authors concluded the diagnostic accuracy was comparable to the studies conducted outside of Africa.

Comments:

1. The accuracy of CT to diagnose AA has been well documented. The authors hypothesized the equipment or experience in interpretation of CT in Tanzania, Africa might be different from the outside of Africa. The images of the study were generated by Philips, Ingenuity model, 128 slices with 64 detector rows. All 4 radiologists interpreting the images had at least trained as Master of Medicine in medical radiology. The equipment and experience in interpretation of CT in this hospital (Aga Khan Hospital) of Tanzania were at the same level as previous studies conducted outside of Africa. The results of this study are expected.

2. This is a retrospective study without clear inclusion criteria. The authors recruited the patients with right lower abdominal pain with suspected AA by physicians (no criteria about how to suspect AA) and underwent CT scans. The inaccurate diagnostic criteria before CT resulted in a high portion of patients without AA in the study. There were 45 case (26%) of AA and 68 cases (39%) of urolithiasis in the study population. The sample size estimation on page 12 used the prevalence of 0.43, which is much higher than 26% in the study population. Therefore, the sample size estimation was invalid.

3. The Fig. 1 illustrated the study population. At top of the Fig. 1, it displayed 7 cases of missing data were excluded. The rest of the Fig. 1 demonstrated the outcomes of the 180 cases. However, there were 7 cases with negative CT and loss of follow-up should be excluded from this study cohort. This exclusion was not demonstrated in the Fig. 1.

Reviewer #2: This is a retrospective diagnostic accuracy study conducted at a private tertiary hospital in Tanzania. The authors aimed to assess the diagnostic accuracy of a computed tomography scan for acute appendicitis in patients with suspected acute appendicitis at the emergency department. They found that the high diagnostic accuracy (97.5%) was comparable to that demonstrated in meta-analysis from studies conducted outside of Africa. This reviewer is an applied statistician and methodologist. I have several concerns.

The entire manuscript needs to be closely edited by someone more proficient with the nuances of the English language as there are multiple sites in which the grammar could be improved.

1. Methodology- The period of this study conducted in the “Methodology” section was different from that recorded in “Abstract”. Which one was correct? Please revise.

2. Methodology- The authors wrote that “Adult patients above ten years of age were included in the study….”. This is different from definition of adult (>14 years of age) in the SRMA conducted by Rud B, et al. (2019). Please explain more.

3. Methodology- Presenting results for diagnostic test assessment include sensitivity, specificity, diagnostic odds ratio, and AUC. The authors should consider adding.

4. Methodology-The authors wrote that “The image interpreters were four consultant medical radiologists rotating through an on-call schedule”. Please give the Kappa value between these four radiologists.

5. Methodology- The authors wrote that “Sample size estimation based on Buderer formula of diagnostic accuracy for unknown disease prevalence”. But this is a retrospective study, why do this calculate? Please explain more.

6. Results-Please give a more detailed demographic table about these 180 included participants.

7. Results- The authors wrote that “the mean participants’ age was 35 years with a range (10 to 71)…”. Formally, for continuous variable like age, it must report as mean with standard deviation or 95% confidence interval. Please revise.

8. Results- Please revise the form for reporting the proportion. As example, line 81 states that “the first assessor was a medical officer in 40 %, but line 83 states that “six percent of the …..”. It must be consistent.

9. Results- Line 96-97 states that “A total of 35 participants determined to have normal appendices, without alternative diagnoses (27) and equivocal CT results (8) ….”. I think that the notations (27) and (8) stood for the numbers of participants but they were same as notations for cite references. Please revise.

10. Results- Please unify the number part, don’t use English and Arabic numerals interchangeably.

11. Results- In Line 108-111, Please give the 95% confidence intervals for sensitivity, specificity, diagnostic accuracy, positive and negative predictive value.

12. Results- The authors wrote that “No significant differences were observed…”. Does any statistical method was used? Which value of p-value was considered to be statistically significant?

13. Can’t find the list of abbreviations.

6. PLOS authors have the option to publish the peer review history of their article (what does this mean?). If published, this will include your full peer review and any attached files.

Reviewer #1: **Yes: **Cheng-Chung Fang

Reviewer #2: No

---

## [Author Response · Author response to Decision Letter 0]

2 Nov 2021

RESPONSE TO REVIEWERS 

Reviewers' comments:

Reviewer's Responses to Questions

Comments to the Author

1. Is the manuscript technically sound, and do the data support the conclusions?

Reviewer #1: Partly

Reviewer #2: Yes

2. Has the statistical analysis been performed appropriately and rigorously? 

Reviewer #1: No

Reviewer #2: No

3. Have the authors made all data underlying the findings in their manuscript fully available?

Reviewer #1: Yes

Reviewer #2: No

4. Is the manuscript presented in an intelligible fashion and written in standard English?

Reviewer #1: Yes

Reviewer #2: No

5. Review Comments to the Author

Reviewer #1: The authors aimed to study the diagnostic accuracy of acute appendicitis (AA) by computed tomography (CT) in Tanzania. They recruited 180 patients suspected AA with CT retrospectively in a single hospital. The reference standards were histological examination of the removed appendix as well as clinical follow‐up for 14 days of participants who did not have surgery. After excluded 7 patients with negative CT and loss of follow-up, 173 participants were included in analysis of which 45 had and 128 did not have AA. The CT's diagnostic accuracy for AA was 97.5%, sensitivity and specificity were 91.6% and 100% respectively. The authors concluded the diagnostic accuracy was comparable to the studies conducted outside of Africa.

Comments:

1. The accuracy of CT to diagnose AA has been well documented. The authors hypothesized the equipment or experience in interpretation of CT in Tanzania, Africa might be different from the outside of Africa. The images of the study were generated by Philips, Ingenuity model, 128 slices with 64 detector rows. All 4 radiologists interpreting the images had at least trained as Master of Medicine in medical radiology. The equipment and experience in interpretation of CT in this hospital (Aga Khan Hospital) of Tanzania were at the same level as previous studies conducted outside of Africa. The results of this study are expected.

There were no prior published studies examining the diagnostic accuracy of CT for acute appendicitis in the sub-Sahara Africa. Given the justification of this study that the assumption of similar performance might not necessarily be true, this question was explored.

2. This is a retrospective study without clear inclusion criteria. The authors recruited the patients with right lower abdominal pain with suspected AA by physicians (no criteria about how to suspect AA) and underwent CT scans. The inaccurate diagnostic criteria before CT resulted in a high portion of patients without AA in the study. There were 45 case (26%) of AA and 68 cases (39%) of urolithiasis in the study population. The sample size estimation on page 12 used the prevalence of 0.43, which is much higher than 26% in the study population. Therefore, the sample size estimation was invalid.

Criteria for suspecting acute appendicitis, right iliac fossa colicky abdominal pain of less than 10 days duration. This has been clarified in the manuscript.

As a result of the lower prevalence of acute appendicitis in this study, the corresponding precision is lowered to 0.067 as opposed to our sample size estimation of 0.05. This information has been added in the manuscript as a limitation of the study.

3. The Fig. 1 illustrated the study population. At top of the Fig. 1, it displayed 7 cases of missing data were excluded. The rest of the Fig. 1 demonstrated the outcomes of the 180 cases. However, there were 7 cases with negative CT and loss of follow-up should be excluded from this study cohort. This exclusion was not demonstrated in the Fig. 1.

Revised.

Reviewer #2: This is a retrospective diagnostic accuracy study conducted at a private tertiary hospital in Tanzania. The authors aimed to assess the diagnostic accuracy of a computed tomography scan for acute appendicitis in patients with suspected acute appendicitis at the emergency department. They found that the high diagnostic accuracy (97.5%) was comparable to that demonstrated in meta-analysis from studies conducted outside of Africa. This reviewer is an applied statistician and methodologist. I have several concerns.

The entire manuscript needs to be closely edited by someone more proficient with the nuances of the English language as there are multiple sites in which the grammar could be improved.

1. Methodology- The period of this study conducted in the “Methodology” section was different from that recorded in “Abstract”. Which one was correct? Please revise.

Revised.

2. Methodology- The authors wrote that “Adult patients above ten years of age were included in the study….”. This is different from definition of adult (>14 years of age) in the SRMA conducted by Rud B, et al. (2019). Please explain more.

In the setting in which the study was conducted, patients above the age of 10 years were managed with adult surgeons. On further review, there were a total of 4 participants below the age of 14, making this 2.2% of the sample size. This information has been added in the manuscript.

3. Methodology- Presenting results for diagnostic test assessment include sensitivity, specificity, diagnostic odds ratio, and AUC. The authors should consider adding.

Considerations given; ROC analysis not deemed necessary as the index test results and gold standard were either positive or negative (dichotomous) and not a scale.

4. Methodology-The authors wrote that “The image interpreters were four consultant medical radiologists rotating through an on-call schedule”. Please give the Kappa value between these four radiologists.

Each radiologist interpreted the results independently, depending on when he/she was on call. This has been clarified in the manuscript.

5. Methodology- The authors wrote that “Sample size estimation based on Buderer formula of diagnostic accuracy for unknown disease prevalence”. But this is a retrospective study, why do this calculate? Please explain more.

Sample size calculation was done to ascertain the confidence of the results and to know the sample size required to reach this confidence.

6. Results-Please give a more detailed demographic table about these 180 included participants.

Additional information provided.

7. Results- The authors wrote that “the mean participants’ age was 35 years with a range (10 to 71)…”. Formally, for continuous variable like age, it must report as mean with standard deviation or 95% confidence interval. Please revise.

Revised.

8. Results- Please revise the form for reporting the proportion. As example, line 81 states that “the first assessor was a medical officer in 40 %, but line 83 states that “six percent of the …..”. It must be consistent.

Revised.

9. Results- Line 96-97 states that “A total of 35 participants determined to have normal appendices, without alternative diagnoses (27) and equivocal CT results (8) ….”. I think that the notations (27) and (8) stood for the numbers of participants but they were same as notations for cite references. Please revise.

Revised.

10. Results- Please unify the number part, don’t use English and Arabic numerals interchangeably.

Revised.

11. Results- In Line 108-111, Please give the 95% confidence intervals for sensitivity, specificity, diagnostic accuracy, positive and negative predictive value.

Revised.

12. Results- The authors wrote that “No significant differences were observed…”. Does any statistical method was used? Which value of p-value was considered to be statistically significant?

This subgroup analysis has been omitted due to limited samples

13. Can’t find the list of abbreviations.

List of Abbreviations Added.

6. PLOS authors have the option to publish the peer review history of their article (what does this mean?). If published, this will include your full peer review and any attached files.

Do you want your identity to be public for this peer review? For information about this choice, including consent withdrawal, please see our Privacy Policy.

Reviewer #1: Yes: Cheng-Chung Fang

Reviewer #2: No

---

## [Decision Letter · Decision Letter 1]

16 Nov 2021

PONE-D-21-13993R1DIAGNOSTIC ACCURACY OF COMPUTED TOMOGRAPHY IN ADULTS WITH SUSPECTED ACUTE APPENDICITIS AT THE EMERGENCY DEPARTMENT IN A PRIVATE TERTIARY HOSPITAL IN TANZANIAPLOS ONE

Dear Dr. Nyamuryekung'e,

Thank you for submitting your manuscript to PLOS ONE. After careful consideration, we feel that it has merit but does not fully meet PLOS ONE’s publication criteria as it currently stands. Therefore, we invite you to submit a revised version of the manuscript that addresses the points raised during the review process. Please revise.

We look forward to receiving your revised manuscript.

Kind regards,

Academic Editor

PLOS ONE

Reviewers' comments:

Reviewer's Responses to Questions

**Comments to the Author**

1. If the authors have adequately addressed your comments raised in a previous round of review and you feel that this manuscript is now acceptable for publication, you may indicate that here to bypass the “Comments to the Author” section, enter your conflict of interest statement in the “Confidential to Editor” section, and submit your "Accept" recommendation.

Reviewer #1: (No Response)

Reviewer #2: (No Response)

2. Is the manuscript technically sound, and do the data support the conclusions?

Reviewer #1: No

Reviewer #2: Partly

3. Has the statistical analysis been performed appropriately and rigorously? 

Reviewer #1: No

Reviewer #2: No

4. Have the authors made all data underlying the findings in their manuscript fully available?

Reviewer #1: Yes

Reviewer #2: No

5. Is the manuscript presented in an intelligible fashion and written in standard English?

Reviewer #1: No

Reviewer #2: No

6. Review Comments to the Author

Reviewer #1: The authors submitted a revised manuscript and responded to the questions. However, my concerns are still unresolved.

1. As the authors’ reply, the inclusion criteria of the study were: patients with right iliac fossa colicky abdominal pain of fewer than 10 days duration. There were no other physical examinations or laboratory tests to justify the impression of acute appendicitis (such as Alvarado score). The loose criteria made the high percentage (39%) of urolithiasis in the study population. The conclusion: “The commonest alternative diagnosis was urolithiasis and should be considered during the evaluation of adult patients with suspected acute appendicitis.” is inappropriate. Emergency physicians usually perform more studies for patients with right lower abdominal colic pain before CT. If we adopt the authors’ criteria, we would perform CT for every patient with right lower abdominal colic pain.

2. The rationale for performing this study is lacking. The reason: “There were no prior published studies examining the diagnostic accuracy of CT for acute appendicitis in the sub-Sahara Africa.” is not acceptable.

Reviewer #2: This is my second review of this manuscript. The authors have replied and made many revisions regarding previous reviewers’ major concerns. Yet there remained some issues.

1. First, the entire manuscript still needs to be closely edited by someone more proficient with the nuances of the English language as there are multiple sites in which the grammar could be improved.

2. Methodology- The authors wrote that “Adult patients above ten years of age were included in the study….”. This is different from the definition of adult (>14 years of age) in the SRMA conducted by Rud B, et al. (2019). Please explain more.

Authors' response: In the setting in which the study was conducted, patients above the age of 10 years were managed with adult surgeons. On further review, there were a total of 4 participants below the age of 14, making this 2.2% of the sample size. This information has been added to the manuscript.

Reviewer response: But I still have concerns about why the author doesn’t include the adult patients above 14 years as same as in the SRMA conducted by Rud, B, et al. (2019).

3. Methodology- Presenting results for diagnostic test assessment include sensitivity, specificity, diagnostic odds ratio, and AUC. The authors should consider adding.

Authors response: Considerations given; ROC analysis not deemed necessary as the index test results and the gold standard were either positive or negative (dichotomous) and not a scale.

Reviewer response: Whether the index test results and the gold standard were either positive or negative (dichotomous), it still can calculate the AUROC.

4. Methodology-The authors wrote that “The image interpreters were four consultant medical radiologists rotating through an on-call schedule”. Please give the Kappa value between these four radiologists.

Authors' response: Each radiologist interpreted the results independently, depending on when he/she was on call. This has been clarified in the manuscript.

Reviewer response: My concern is that whether the image interpreters were done by two independent radiologists?

5. Methodology- The authors wrote that “Sample size estimation based on Buderer formula of diagnostic accuracy for unknown disease prevalence”. But this is a retrospective study, why does this calculate? Please explain more.

Authors' response: Sample size calculation was done to ascertain the confidence of the results and to know the sample size required to reach this confidence.

Reviewer response: To my knowledge, more data is better. Additionally, when the authors do sample size estimation based on the Buderer formula, the disease prevalence that the authors utilized according to the SRMA is 0.43 as described in Line 70-73, but the prevalence of this study is 0.25. Would it cause underestimation or upper estimation of the power?

6. Results-Please give a more detailed demographic table about these 180 included participants.

Authors response: Additional information provided.

Reviewer response: I still can’t find the additional information in one Table. As the authors describe the summary results through Line 81 to 97 in the Result section.

7. Results- The authors wrote that “No significant differences were observed…”. Does any statistical method was used? Which value of p-value was considered to be statistically significant?

Authors response: This subgroup analysis has been omitted due to limited samples

Reviewer response: Tests for Two Independent Sensitivities (netdna-ssl.com)

8. In Abstract, the Confidence Interval of 91.96 to 100%), 96.7% (95% Confidence Interval of 91.82 to 99.100%), and 97.6% (95% Confidence Interval of 93.88 to 99.31%), respectively. What is 99.100%? Please revised. Additionally, please change (95% Confidence Interval of 91.82 to 99.100%) to (95% C.I.: 91.82-99) through all manuscript.

9. In Result, the authors wrote that “Participants above 14 years of age contributed to 97.8% of 84 the sample; there were 2 participants of age 12 years and 1 participant each of age 11 and 10 years”. I really can’t understand and what it means?

10. In Line 110-112, the authors wrote that “A total of 35 participants, of which 27 had determined to have normal appendices without 111 alternative diagnoses and 8 with equivocal features of AA on CT”. but the total included participants is 178, why 35?

7. PLOS authors have the option to publish the peer review history of their article (what does this mean?). If published, this will include your full peer review and any attached files.

Reviewer #1: **Yes: **Cheng-Chung Fang

Reviewer #2: No

---

## [Author Response · Author response to Decision Letter 1]

31 Dec 2021

Response to Reviewer #1: 

1. Details of evaluations performed by emergency medicine physicians including additional tests have been included in the manuscript.

2. The rationale has been refined further.

Responses to Reviewer #2: 

1. Grammatical and typographical errors reviewed and rectified. 

2. To conform with the SMRA study, patients below the age of 14 have been excluded from the analysis of this study.

3. AUROC added and additional subgroup analyses.

4. The images were interpreted by one of four radiologists who was on call. This information has been clarified in the manuscript.

5. More data is certainly better. However, this also means more use of resources for data collection procedures, and sometimes this has ethical implications, hence calculating what would be needed allows one to know how much data collection is adequate. The lower prevalence of acute appendicitis in our study affected the precision of the results. This has been discussed in the discussion section. 

6. Table one has been reviewed to contain more demographic details.

7. The reporting of results has been revised.

9. This section has been revised.

10. This section has been revised.

---

## [Decision Letter · Decision Letter 2]

12 Jan 2022

PONE-D-21-13993R2DIAGNOSTIC ACCURACY OF COMPUTED TOMOGRAPHY IN ADULTS WITH SUSPECTED ACUTE APPENDICITIS AT THE EMERGENCY DEPARTMENT IN A PRIVATE TERTIARY HOSPITAL IN TANZANIAPLOS ONE

Dear Dr. Nyamuryekung'e,

Thank you for submitting your manuscript to PLOS ONE. After careful consideration, we feel that it has merit but does not fully meet PLOS ONE’s publication criteria as it currently stands. Therefore, we invite you to submit a revised version of the manuscript that addresses the points raised during the review process.

Please revise.

We look forward to receiving your revised manuscript.

Kind regards,

Academic Editor

PLOS ONE

Reviewers' comments:

Reviewer's Responses to Questions

**Comments to the Author**

1. If the authors have adequately addressed your comments raised in a previous round of review and you feel that this manuscript is now acceptable for publication, you may indicate that here to bypass the “Comments to the Author” section, enter your conflict of interest statement in the “Confidential to Editor” section, and submit your "Accept" recommendation.

Reviewer #1: (No Response)

Reviewer #2: (No Response)

2. Is the manuscript technically sound, and do the data support the conclusions?

Reviewer #1: No

Reviewer #2: (No Response)

3. Has the statistical analysis been performed appropriately and rigorously? 

Reviewer #1: No

Reviewer #2: (No Response)

4. Have the authors made all data underlying the findings in their manuscript fully available?

Reviewer #1: No

Reviewer #2: (No Response)

5. Is the manuscript presented in an intelligible fashion and written in standard English?

Reviewer #1: No

Reviewer #2: (No Response)

6. Review Comments to the Author

Reviewer #1: The authors submitted a revised manuscript and responded to the questions. The authors stated that “The Alvarado score (AS) was not documented. However, based on symptoms assessed and the results of the CBC, the AS was computed for each patient. The mean AS was 4.64 (95% CI 4.32-4.97), those with AS of 1 to 4 were 50.6%”. As the authors’ replied, the inclusion criteria of the study were too loose. The AS of the patients less than 4 were 50.6% of the study patients, who should not receive abdominal CT for suspect acute appendicitis. The loose criteria made the high percentage (39%) of urolithiasis in the study population. The conclusion: “The commonest alternative diagnosis was urolithiasis and should be considered during the evaluation of adult patients with suspected acute appendicitis.” is inappropriate. Emergency physicians usually perform more examinations, such as urinalysis and abdominal ultrasonography for patients with right lower abdominal colic pain before CT. The diagnosis of urolithiasis is easy to identify by these examinations. If we adopt the authors’ criteria, we would perform CT for every patient with right lower abdominal colic pain.

Reviewer #2: This is my second review of this manuscript. The authors have replied and made revisions regarding previous reviewers’ major concerns. Yet there remained more issues.

1. In my first comment, the authors stated that “The image interpreters were four consultant medical radiologists rotating through an on-call schedule. Please give the Kappa value between these four radiologists”. And the authors replied as “Each radiologist interpreted the results independently, depending on when he/she was on call. This has been clarified in the manuscript”. For this comment, my question was “whether the same image was interpreted by two or more radiologists or just one? If two or more radiologists have interpreted the same image, I would like to know how the degree of the consistent outcome through giving the kappa values? But if just one radiologist interpreted for one image, this may be lead to an inconsistent outcome and must have been addressed.

2. In my first comment, I have stated that “Please give the 95% confidence intervals for sensitivity, specificity, diagnostic accuracy, positive and negative predictive value”, but I suggest the authors revise the terms of confidence intervals into more formula terms. For example, (95% Confidence Interval of 0.79 to 0.96) to (95% C.I.: 0.79-0.96). the same terms through this manuscript like this term also please address accordingly. Additionally, do not use the percentage and decimal point interchangeably, please unify.

7. PLOS authors have the option to publish the peer review history of their article (what does this mean?). If published, this will include your full peer review and any attached files.

Reviewer #1: No

Reviewer #2: No

---

## [Author Response · Author response to Decision Letter 2]

19 Mar 2022

Comments to the Author

Reviewer #1: 

The authors submitted a revised manuscript and responded to the questions. The authors stated that “The Alvarado score (AS) was not documented. However, based on symptoms assessed and the results of the CBC, the AS was computed for each patient. The mean AS was 4.64 (95% CI 4.32-4.97), those with AS of 1 to 4 were 50.6%”. As the authors’ replied, the inclusion criteria of the study were too loose. The AS of the patients less than 4 were 50.6% of the study patients, who should not receive abdominal CT for suspect acute appendicitis. The loose criteria made the high percentage (39%) of urolithiasis in the study population. The conclusion: “The commonest alternative diagnosis was urolithiasis and should be considered during the evaluation of adult patients with suspected acute appendicitis.” is inappropriate. Emergency physicians usually perform more examinations, such as urinalysis and abdominal ultrasonography for patients with right lower abdominal colic pain before CT. The diagnosis of urolithiasis is easy to identify by these examinations. If we adopt the authors’ criteria, we would perform CT for every patient with right lower abdominal colic pain.

Response: 

Thank you for your comments. For uniformity, the inclusion criteria were adopted from a meta-analysis and systematic on the same subject by Bo Rud et al, 2019. The use of USS prior to CT introduces a confounding effect on the assessment of CT’s diagnostic performance, and this is an exclusion criterion, in this study and the systematic review quoted above. 

The concerns on the use of AS have been analyzed and discussed, using the AS threshold of 4 would have resulted in 50% less CT scans, and missed 4 AA. Recommendations for the use of AS in assessing patients with suspected acute appendicitis have been emphasized. 

The recommendation by the authors referred to above had been rectified in an earlier revised version and is not present in the current revised version. Results of urinalysis done have been described and analyzed.

Reviewer #2: 

This is my second review of this manuscript. The authors have replied and made revisions regarding previous reviewers’ major concerns. Yet there remained more issues.

1. In my first comment, the authors stated that “The image interpreters were four consultant medical radiologists rotating through an on-call schedule. Please give the Kappa value between these four radiologists”. And the authors replied as “Each radiologist interpreted the results independently, depending on when he/she was on call. This has been clarified in the manuscript”. For this comment, my question was “whether the same image was interpreted by two or more radiologists or just one? If two or more radiologists have interpreted the same image, I would like to know how the degree of the consistent outcome through giving the kappa values? But if just one radiologist interpreted for one image, this may be lead to an inconsistent outcome and must have been addressed.

2. In my first comment, I have stated that “Please give the 95% confidence intervals for sensitivity, specificity, diagnostic accuracy, positive and negative predictive value”, but I suggest the authors revise the terms of confidence intervals into more formula terms. For example, (95% Confidence Interval of 0.79 to 0.96) to (95% C.I.: 0.79-0.96). the same terms through this manuscript like this term also please address accordingly. Additionally, do not use the percentage and decimal point interchangeably, please unify.

Response to comment 1.

The concern for four independent radiologists evaluating the CT introduces inter-observer variability. This is has been discussed as a limitation of this study.

Response to comment 2.

This has been rectified. Thank you for your comments.

---

## [Decision Letter · Decision Letter 3]

12 May 2022

PONE-D-21-13993R3DIAGNOSTIC ACCURACY OF COMPUTED TOMOGRAPHY IN ADULTS WITH SUSPECTED ACUTE APPENDICITIS AT THE EMERGENCY DEPARTMENT IN A PRIVATE TERTIARY HOSPITAL IN TANZANIAPLOS ONE

Dear Dr. Nyamuryekung'e,

Thank you for submitting your manuscript to PLOS ONE. After careful consideration, we feel that it has merit but does not fully meet PLOS ONE’s publication criteria as it currently stands. Therefore, we invite you to submit a revised version of the manuscript that addresses the points raised during the review process.

Please revise.

We look forward to receiving your revised manuscript.

Kind regards,

Academic Editor

PLOS ONE

Journal Requirements:

Reviewers' comments:

Reviewer's Responses to Questions

**Comments to the Author**

1. If the authors have adequately addressed your comments raised in a previous round of review and you feel that this manuscript is now acceptable for publication, you may indicate that here to bypass the “Comments to the Author” section, enter your conflict of interest statement in the “Confidential to Editor” section, and submit your "Accept" recommendation.

Reviewer #1: (No Response)

Reviewer #2: All comments have been addressed

Reviewer #3: All comments have been addressed

Reviewer #4: (No Response)

2. Is the manuscript technically sound, and do the data support the conclusions?

Reviewer #1: No

Reviewer #2: Yes

Reviewer #3: Yes

Reviewer #4: Yes

3. Has the statistical analysis been performed appropriately and rigorously? 

Reviewer #1: No

Reviewer #2: Yes

Reviewer #3: Yes

Reviewer #4: Yes

4. Have the authors made all data underlying the findings in their manuscript fully available?

Reviewer #1: No

Reviewer #2: No

Reviewer #3: Yes

Reviewer #4: No

5. Is the manuscript presented in an intelligible fashion and written in standard English?

Reviewer #1: No

Reviewer #2: Yes

Reviewer #3: Yes

Reviewer #4: Yes

6. Review Comments to the Author

Reviewer #1: The authors submitted the third revision of their manuscript. Same as the previous revisions, the authors did not directly respond to my questions, and they did not make point-to-point responses. The study population did not change after these revisions. Therefore, my questions were not resolved after the revisions. This study is a retrospective chart review study and has many inclusion biases when calculating the diagnostic accuracy of CT in acute appendicitis. I do not wish to make comments anymore, as my previous comments are still unresolved.

Reviewer #2: This is my third review of this manuscript. The authors have replied and made many revisions regarding previous reviewers’ major concerns. I have no more comments. Thanks.

Reviewer #3: Page 12, line 128: In the discussion section, the statement of "10239 hours" is incorrect, and should be "10239 minutes".

Reviewer #4: Dear authors,

Please see my comments in the attached pdf-file. You revised the manuscript several times before so I have focused on only important questions. I wish you the very best. Good Luck!

7. PLOS authors have the option to publish the peer review history of their article (what does this mean?). If published, this will include your full peer review and any attached files.

Reviewer #1: No

Reviewer #2: No

Reviewer #3: **Yes: **JyhWen Chai

Reviewer #4: No

---

## [Author Response · Author response to Decision Letter 3]

29 Sep 2022

Response to Reviewers

Reviewer #1: The authors submitted the third revision of their manuscript. Same as the previous revisions, the authors did not directly respond to my questions, and they did not make point-to-point responses. The study population did not change after these revisions. Therefore, my questions were not resolved after the revisions. This study is a retrospective chart review study and has many inclusion biases when calculating the diagnostic accuracy of CT in acute appendicitis. I do not wish to make comments anymore, as my previous comments are still unresolved.

Response to reviewer 1: Thank you for your review. The risk of inclusion bias due to the retrospective nature of the study has been added as a limitation. 

Below are the comments that were made on the previous review, and I have tried to break them down so that I may be able to provide appropriate responses

Reviewer #1: 

The authors submitted a revised manuscript and responded to the questions. The authors stated that “The Alvarado score (AS) was not documented. However, based on symptoms assessed and the results of the CBC, the AS was computed for each patient. 

1. The mean AS was 4.64 (95% CI 4.32-4.97), those with AS of 1 to 4 were 50.6%”. As the authors’ replied, the inclusion criteria of the study were too loose. The AS of the patients less than 4 were 50.6% of the study patients, who should not receive abdominal CT for suspect acute appendicitis. The loose criteria made the high percentage (39%) of urolithiasis in the study population. 

2. The conclusion: “The commonest alternative diagnosis was urolithiasis and should be considered during the evaluation of adult patients with suspected acute appendicitis.” is inappropriate. 

3. Emergency physicians usually perform more examinations, such as urinalysis and abdominal ultrasonography for patients with right lower abdominal colic pain before CT. The diagnosis of urolithiasis is easy to identify by these examinations. 

4. If we adopt the authors’ criteria, we would perform CT for every patient with right lower abdominal colic pain.

Response to Reviewer 1: 

1. For uniformity, the inclusion criteria were adopted from a meta-analysis and systematic on the same subject by Bo Rud et al, 2019, whereby the Alvarado Score was not used for participant selection. However, as noted Alvarado Score was not performed in routine clinical practice to evaluate patients with suspected Acute Appendicitis; it was calculated in this study based on the workup performed. 

Had the AS been conducted in routine clinical practice and an AS of < 4 chosen to determine the need to perform CT, 50% fewer CT scans would have been conducted and missed 4 Acute Appendicitis. In the discussion, based on these results, we emphasise and advocate the routine use of AS to guide further management of patients with suspected AA.

2. This conclusion was removed in an earlier version and is not present in the current version.

3. Patients who underwent USS before CT were excluded from this study as this would introduce a confounding effect on CT’s diagnostic performance. Urinalysis was done in 134 patients and was normal in 79.1%, 19.4% showing microscopic hematuria, and pyuria with leucocytes in 1.5%. Urinalysis was not associated with a diagnosis of AA on binomial logistic regression (Positive for Acute Appendicitis or negative for Acute Appendicitis). This information is included in the revised manuscript 

4. In the discussion section, we have emphasised and advocated the need to conduct an AS for patients with suspected AA before further management, as the results of this study also show that it would have decreased unnecessary CT scans.

Reviewer #2: This is my third review of this manuscript. The authors have replied and made many revisions regarding previous reviewers’ major concerns. I have no more comments. Thanks.

Response to reviewer 2: Thank you for your review.

Reviewer #3: Page 12, line 128: In the discussion section, the statement of "10239 hours" is incorrect, and should be "10239 minutes".

Response to reviewer 3: Thank you for the review. The change has been made

Reviewer #4: Dear authors, please see my comments in the attached pdf-file. You revised the manuscript several times before so I have focused on only important questions. I wish you the very best. Good Luck!

Response to reviewer 4: Thank you for the comments. 

1. It is not unusual for patients to present late in this setting present due to various health-seeking behaviours, and we wanted to ensure we do not exclude them from the study due to health-seeking behaviour. However, by day 5, the cumulative percentage of patients who had presented to the EMD was 93%. 

2. A medical officer is a registered medical doctor with a Doctor of Medicine degree qualification only. This definition has been added to the revised manuscript

3. A general surgery specialist was the first assessor in 23% of the cases. This has been included in the revised manuscript

4. We have rectified the noted errors in the placement of less than and greater than symbols.

5. Duration of illness was defined as the number of days from onset of abdominal pain to emergency medicine department orientation. The definition has been included in the revision.

---

## [Decision Letter · Decision Letter 4]

13 Oct 2022

DIAGNOSTIC ACCURACY OF COMPUTED TOMOGRAPHY IN ADULTS WITH SUSPECTED ACUTE APPENDICITIS AT THE EMERGENCY DEPARTMENT IN A PRIVATE TERTIARY HOSPITAL IN TANZANIA

PONE-D-21-13993R4

Dear Dr. Nyamuryekung'e,

We’re pleased to inform you that your manuscript has been judged scientifically suitable for publication and will be formally accepted for publication once it meets all outstanding technical requirements.

Kind regards,

Academic Editor

PLOS ONE

Additional Editor Comments (optional):

Reviewers' comments:

Reviewer's Responses to Questions

**Comments to the Author**

1. If the authors have adequately addressed your comments raised in a previous round of review and you feel that this manuscript is now acceptable for publication, you may indicate that here to bypass the “Comments to the Author” section, enter your conflict of interest statement in the “Confidential to Editor” section, and submit your "Accept" recommendation.

Reviewer #2: All comments have been addressed

Reviewer #4: All comments have been addressed

2. Is the manuscript technically sound, and do the data support the conclusions?

Reviewer #2: Yes

Reviewer #4: Yes

3. Has the statistical analysis been performed appropriately and rigorously? 

Reviewer #2: Yes

Reviewer #4: Yes

4. Have the authors made all data underlying the findings in their manuscript fully available?

Reviewer #2: No

Reviewer #4: Yes

5. Is the manuscript presented in an intelligible fashion and written in standard English?

Reviewer #2: Yes

Reviewer #4: Yes

6. Review Comments to the Author

Reviewer #2: This is my fourth review of this manuscript. The authors have replied and made many

revisions regarding previous reviewers’ major concerns. I have no more comments. Thanks.

Reviewer #4: Dear authors, thank you for your reply. I am very happy with the revision. You have answered all my comments.

7. PLOS authors have the option to publish the peer review history of their article (what does this mean?). If published, this will include your full peer review and any attached files.

Reviewer #2: No

Reviewer #4: No

---

## [Editor Report · Acceptance letter]

19 Oct 2022

PONE-D-21-13993R4 

Diagnostic accuracy of computed tomography in adults with suspected acute appendicitis at the emergency department in a private tertiary hospital in Tanzania 

Dear Dr. Nyamuryekung'e:

I'm pleased to inform you that your manuscript has been deemed suitable for publication in PLOS ONE. Congratulations! Your manuscript is now with our production department. 

Kind regards, 

on behalf of

Dr. Robert Jeenchen Chen 

Academic Editor

PLOS ONE